# The Effect of Type of Vegetable Fat and Addition of Antioxidant Components on the Physicochemical Properties of a Pea-Based Meat Analogue

**DOI:** 10.3390/foods13010071

**Published:** 2023-12-24

**Authors:** Klaudia Kołodziejczak, Anna Onopiuk, Arkadiusz Szpicer, Andrzej Poltorak

**Affiliations:** Department of Technique and Food Development, Institute of Human Nutrition Sciences, Warsaw University of Life Sciences, Nowoursynowska 159c Street, 32, 02-776 Warsaw, Poland; klaudia_kolodziejczak@sggw.edu.pl (K.K.); arkadiusz_szpicer@sggw.edu.pl (A.S.); andrzej_poltorak@sggw.edu.pl (A.P.)

**Keywords:** meat substitutes, antioxidant activity, anti-inflammatory activity, flavonoids, polyphenolic compounds, pea protein

## Abstract

In recent years, interest in functional foods and meat analogues has increased. This study investigated the effect of the type of vegetable fat and ingredients with antioxidant activity on the properties of a meat analogue based on textured pea protein. The possibility of using acai oil (AO), canola oil (CO) and olive oil (OO); propolis extract (P); buckwheat honey (H); and jalapeno pepper extract (JE) was investigated. The texture, colour and selected chemical parameters of plant-based burgers were analysed. Results showed that burgers from control group had the lowest hardness, while burgers with honey had the highest. The highest MUFA content was found in samples with olive oil. Samples with honey were characterised by the highest content of polyphenols, flavonoids and antioxidant capacity. The highest overall acceptability was observed in burgers from the JE-CO group. Therefore, it is possible to use selected ingredients with antioxidant activity in the recipe for a plant-based burger with high product acceptability.

## 1. Introduction

Reducing or excluding meat from the diet is a decision made by consumers as a result of growing dietary awareness, concern for the environment, and the availability of key resources, biodiversity, and animal welfare [1]. Important factors in reducing meat consumption also include consumer concerns about the influence of meat and processed meat products consumption on the risk of diet-related diseases [2], as well as the epidemiological risk of zoonotic diseases and the increasing antibiotic resilience of pathogenic microorganisms [3,4]. The environmental impact of livestock production, with its relatively low efficiency and high consumption of natural resources compared to plant production and the growing global human population, indicates the need to investigate alternative sources of protein in the human diet [3,5,6]. Simultaneously, the significant role of meat in the human diet cannot be neglected, particularly in terms of providing highly digestible and complete protein [7].

In recent years, the percentage of people on vegan, vegetarian, and flexitarian diets has been increasing [8]. The assumptions of this type of diet include the total or partial exclusion of animal products. Therefore, the proper balance of the diet requires protein from other sources. The most common are plant proteins, especially those derived from legumes. One form of plant protein consumption highly valued by consumers, especially in the initial stage of conversion to a plant-based diet, is meat analogues. These products should be characterised by sensory properties and nutritional value similar to meat and meat products, so they could play a function identical to meat in daily meals [9,10]. That is a significant simplification for consumers, since the wide variety of meat analogues available on the market makes it possible to directly replace meat with its analogue in almost any type of dish [11].

The key ingredient in a meat analogue is protein. Even though almost all plant proteins can become the basis of a meat analogue, some factors determine their suitability for this application. Animal protein is characterised by high digestibility, full aminogram, and specific textural properties. In contrast, most plant proteins have a limiting amino acid and different functional properties. Among plant sources, soy (about 1.00), canola, potato, pea, and quinoa protein (value of at least 0.75) have the highest PDCAAS [7]. In the production of meat analogues, soybean and wheat proteins are widely used [12]. An important aspect is their high availability and relatively low price. However, soy and wheat are raw materials with high allergenic potential, and some consumers demonstrate a low level of tolerance for consuming soy- or gluten-containing products [13]. Recently, researchers have investigated the possibility of using proteins from other legumes, fungal fermentation, insects, microalgae, or bacteria to produce meat analogues [4]. Pea protein is the most promising plant protein for producing meat analogues in terms of consumer acceptance, availability, low allergenicity, and technological properties such as water- and oil-holding capacity, gelation, and solubility. This protein is valuable in terms of nutritional value, high lysine and threonine content and low glycaemic index [14,15]. The technological properties of plant proteins are highly variable and depend, among other things, on the method of protein isolation [15]. They can be modified to some extent through the use of appropriate chemical, physical or enzymatic processing [16].

Considering its significant influence on the formation of nutritional value and sensory properties, plant fat is one of the most essential components of a meat analogue, in addition to protein. Plant fat has a texturing role and carries flavour. It also functions as a carrier of fat-soluble vitamins, which can be introduced into the product as a fortification or with other ingredients. Canola oil and sunflower oil are widely applied. Solid fats extracted from coconut and cocoa are used to achieve a marbling effect, while other fats (e.g., sesame oil, avocado oil) are used to improve the fatty acid profile and flavour of the product [17,18,19]. Many other ingredients create the formulation of the meat analogue depending on the expected characteristics of the product, including pectins, polysaccharide gums, extracts with colouring activity, herbs, spices, yeast extract, nucleotides, sugar, enzymes, and antioxidants [12,18,19,20].

In the production of meat analogues, numerous technological processes are used. Due to the cost and efficiency of the process, low- (<35%) and high- (>50%) moisture extrusion are the two most broadly used processes. However, new innovative techniques are also being developed and researched, e.g., wet-spinning, electrospinning, 3D printing and shear cell technique [9,11,21,22]. Although necessary to obtain the expected textural properties, intensive processing carries the risk of unfavourable transformations, including oxidative reactions. For this reason, several additives are used to protect the product and maintain its quality. The presence of such substances, such as those of synthetic origin, is perceived negatively by consumers who expect the so-called “clean label” [23]. Antioxidants of natural origin may be the answer to consumer expectations in this area. These include polyphenols and essential oils used as a product ingredient or packaging component. The possibility of using such ingredients as extracts from pepper (*Piper nigrum* L.), oregano (*Origanum vulgare* L.) juniper, (*Juniperus communis* L.), and jalapeño extract, as well as catuaba, galangal, and honey to extend the shelf life of meat products has been studied [24,25,26,27].

Consumer nutritional awareness is increasing simultaneously with the standards for food products including meat analogues. Desirable sensory properties, “clean label”, and natural composition are all expected, while adding health-promoting ingredients is an especially appreciated value. This study aimed to investigate the effect of the type of plant fat used on the physicochemical properties of the meat analogue. Selected ingredients with antioxidant activity were used to formulate plant-based burgers. The texture, colour, fatty acid profile, total antioxidant capacity and anti-inflammatory properties, polyphenol and flavonoid content, and consumer acceptability of the plant burger based on textured pea protein were analysed.

## 2. Materials and Methods

### 2.1. Sample Preparation and Frying Procedure

For the preparation of the burgers, the following ingredients were used: textured pea protein (Ingredion, UK); canola oil (Zlote Łany, Poland); olive oil (Borges, Spain); acai oil (PuroPura, Brazil); propolis (Pasieka Kaszubska, Poland); buckwheat honey (Pasieka Kaszubska, Poland); jalapeno extract (Result, Poland); carrageenan (Agnex, Poland); amidated pectin (Rafex, Poland); cep flavour (Bellako, Poland); yeast extract (Bellako, Poland); beet juice powder (Bellako, Poland); dried onion (Planteon, Poland); and spices, namely sweet paprika, smoked paprika, hot paprika, and bear garlic (Kotany, Poland). Detailed recipes and the division into research groups are shown in Table 1.

A meat analogue in the form of a burger based on pea protein was prepared by adding spices, beet juice, yeast extract, and mushroom flavouring to water. Jalapeno extract, propolis extract, and honey were added with water in groups of H-CO, H-OO, H-AO, H-CO, P-OO, P-AO, JE-CO, JE-OO, JE-AO. Then, 75% of the prepared mixture was used to hydrate the textured pea protein for 15 min at ambient temperature. Then dried onions, plant fat (depending on the research group: canola oil (CO), olive oil (OO) or acai oil (AO)), carrageenan, and pectin dissolved in the remaining volume of water were added. All ingredients were mixed carefully and subsequently formed into a burger shape with a diameter of approximately 10 ± 0.1 cm, a height of 1.5 ± 0.1 cm, and a weight of 120 ± 5 g. The burgers were weighed and then heat-treated in a grill pan until the geometric centre of the product reached 75 °C (about 5 min). The cooled burgers were weighed. A total of 6 burgers from each group were used to perform a semi-consumer evaluation. The remaining burgers were used to perform colour and texture analysis. In addition, samples were secured for further testing (vacuum packing in polyethylene bags).

### 2.2. Colour Measurement

The colour analysis of the burgers was performed in the CIE L* a* b* system using the reflection method with a Minolta CR-400 colourimeter (Konica Minolta Inc., Tokyo, Japan). The instrument was previously calibrated with a white standard (L* = 98.45, a* = −0.10, b* = −0.13). The colour of raw and heat-treated burgers was measured on the 1st and 6th days of storage. The following parameters were measured: L* (lightness), a* (from −a* “greenness” to +a* “redness”) and b* (from −b* “blueness” to +b* “yellowness”). The results are presented as the arithmetic mean of 10 measurements.

### 2.3. Texture Measurement

The following texture parameters of meat analogues were measured: springiness (the ratio of the height reached after the first compression to the basic sample height), cohesiveness (proportion of the area under the curve from the second to the first compression) and hardness (the maximum force of the 1st compression; N/cm^2^). An Instron 5965 universal testing machine (Instron, Norwood, MA, USA) was used for this purpose. A double compression test was applied in which the sample was compressed to 50% of its height. The relaxation time was 10 s, and the head speed was 10 mm/min. Measurements were taken from 6 repetitions on cylindrical samples with a height of 1.5 cm and a diameter of 2.5 cm.

### 2.4. Fatty Acid Profile Analysis

For fatty acid profile analysis, 16 g samples were weighed into glass screw-top bottles and mixed with a 100 mL petroleum ether:acetone (2:1) mixture. The samples were intensively shaken on a shaker (MyLab SLRM-3, NanoEnTek Inc., Seoul, Republic of Korea) and then homogenised. In the next step, the homogenisate was filtered through Whatman No. 1 filter paper into glass separators. The bottle was rinsed with approximately 10 mL of petroleum ether–acetone mixture. A 1 mL quantity of saturated salt solution was added and shaken for 30 s. Then 1 mL of distilled water was added and shaken again. The lower layer from the separator was poured into a glass beaker, and the upper layer was transferred through a filter with anhydrous sodium sulphate to a glass ground flask. The solvent was evaporated under reduced pressure on a slow-speed evaporator (ROTAVAPOR R-100 Büchi, Switzerland). The flask was weighed, and the fat was dissolved in hexane in an amount equivalent to the ratio of 1 mL of hexane:100 mg of fat. A 2 mL quantity of the hexane solution was transferred to glass bottles, 2 mL of 1 M KOH was added and incubated in a water bath with shaking for 6 min at 55 °C. Then 2 mL of distilled water, 1 mL of salt water, and 2 mL of hexane were added to the samples and left for 40 min in a darkened place until phase separation. The clear layer was transferred to bottles with anhydrous sodium VI sulphate. The liquid from above the deposit was then transferred to an autosampler vial. The fatty acid profile of the samples was determined by gas chromatography. For this purpose, a Shimadzu GC-2010 gas chromatograph with a flame ionisation detector (FID) equipped with an RT^®^ 2560 silica column (100 m × 0.25 mm ID and 0.2 µm film thickness) (RESTEK, Bellefonte, PA, USA) was used. The analysis was performed in 3 replicates. Fatty acids were identified by comparing the retention times observed with those of fatty acid standards (SupelcoTM 37 Component FAME mix, Sigma, St. Louis, MO, USA). The results were presented as an arithmetic mean and expressed in grams per 100 g of fat.

### 2.5. Determination of Phenolic Compounds

#### 2.5.1. Total Phenolic Content (TPC)

The total phenolic compound content (TPC) was determined according to the methodology described by Singleton and Rossi [28]. In the first step, 2.5 g of the sample was homogenised (Ultra Turrax, IKA T18 basic, Berlin, Germany) with 7.5 mL of ethanol. The extract was filtered through filter paper, and 0.1 mL of the filtrate was mixed with 6.0 mL of distilled water and 0.5 mL of Folin and Ciocalteu phenol reagent. After 3 min, 1.5 mL of sodium carbonate (7.5% *w*/*v*) was added. The mixture was incubated for 40 min in a water bath (WNB 7 Memmert, Germany) at 40 °C. Absorbance was measured spectrophotometrically at λ = 760 nm (Tecan Spark™ 10 M, Männedorf, Switzerland). Total phenolic content was expressed as the gallic acid equivalent (GA) based on a standard curve. Results are presented as the arithmetic mean of three replicates in mg GA per 100 g sample.

#### 2.5.2. Total Flavonoid Content (TFC)

Total flavonoid content (TFC) was determined using spectrophotometric measurement. One mL of 2% water solution of AlCl3-6H2O was added to 1 mL of ethanol extract of the sample (prepared analogously to Section 2.5.1) and mixed intensively. The samples were incubated at ambient temperature for 10 min. At the same time, a blank sample without extract was prepared. Absorbance was measured at 420 nm (Tecan Spark™ 10 M, Männedorf, Switzerland). The total flavonoid content was determined from the standard curve for quercetin. The results were expressed as the arithmetic mean of the three measurements in mg of quercetin per 100 g sample.

### 2.6. Anti-Inflammatory Compounds

The anti-inflammatory properties of the tested burgers were analysed using the method described by the Osés et al. [29]. A 5 g quantity of the sample was weighed and homogenised with 10 mL of 50% aqueous ethanol solution (Ultra Turrax homogeniser, T18 basic, IKA, Germany). Then 70 μL of 5 mg/mL hyaluronic acid sodium salt from Streptococcus equi and 100 μL of buffer (0.2 M sodium formate, 0.1 M sodium chloride and 0.2 mg/mL bovine serum albumin, pH adjusted to 3.68 with formic acid) were added to 200 μL of the sample. The mixture was carefully mixed and incubated for 10 min at 37 °C (water bath, WNB 7, Memmert, Schwabach, Germany). Then 50 μL of hyaluronidase from bovine testes type IV-S (600 U/mL) prepared in 0.9% sodium chloride was added to the sample and incubated for another 60 min. The enzymatic reaction was stopped by adding 100 μL of 0.8 M potassium tetraborate to the sample. After another 3 min of incubation, the samples were removed from the bath and cooled to room temperature. Then 750 μL of dimethylaminobenzaldehyde (DMAB) solution was added (2 g of DMAB was added to 2.5 mL of 10 N hydrochloric acid and 17.5 mL of glacial acetic acid, and then a 50% solution with glacial acetic acid was prepared). Samples that were prepared in this way were incubated for 20 min, and then the absorbance was measured spectrophotometrically at 586 nm (SparkTM 10 M, Tecan Group Ltd., Männedorf, Switzerland). The blank sample was buffered, and a standard curve was prepared using an N-acetyl-d-glucosamine standard solution. Measurements were carried out in triplicate. One unit (1 U) of hyaluronidase activity catalyses the release of 1 μmol N-acetyl-d-glucosamine (NAG) per min under specified conditions. Inhibition of the enzyme (%) was calculated from the following Equation (1):I = (1 − Y/X) × 100(1)
where I (%)–percentage of inhibition, X–μmol of NAG in the control sample and Y–μmol of NAG of each sample reaction.

### 2.7. DPPH Radical Scavenging Activity

The total antioxidant capacity of heat-treated burgers was measured. For this purpose, the free radical scavenging capacity of DPPH (1,1-diphenyl-2-picrylhydrazyl) was analysed based on the procedure by Sánchez-Moreno et al. [30]. In the first step of the analysis, 2.5 g of the sample was weighed and then homogenised with 7.5 mL of ethanol for 2 min at 7500 rpm (Ultra Turrax homogeniser, T18 basic, IKA Werke, Staufen, Germany). The samples were extracted for 10 min at room temperature using a shaker (MyLab SLRM-3, NanoEnTek Inc., Seoul, Republic of Korea). The samples were then centrifuged at 1800 rpm for 10 min (MPW Med. Instruments, Warszawa, Poland). A 0.5 mL quantity of liquid from above the precipitate was mixed with 3.5 mL of ethanolic 0.1 mM DPPH solution. The samples were vortexed and stored in a dark place at 25 °C for 20 min. A wavelength of 510 nm (SparkTM 10M, Tecan Group Ltd., Männedorf, Switzerland) was used for spectrophotometric absorbance measurement. Ethanol was used as a blank. Measurements were made in triplicate, and the result of total antioxidant capacity is presented as the arithmetic mean of the values calculated according to Equation (2):TAA (%) = (1− As/Ac) × 100(2)
where TAA (%)—reduction of DPPH, As is the absorbance of the test sample and Ac is the absorbance of the control (containing all the reagents except extract).

### 2.8. Semi-Consumer Evaluation of Plant-Based Meat Analogues

The semi-consumer acceptability of plant-based pea burgers was evaluated by 34 consumers (men and women aged 25 to 55). The study was conducted under standard conditions, at room temperature and under natural light. Samples in the form of half a burger on paper plates were presented to the participants. The samples were randomly positioned and coded with random numbers. The participants’ role was to place a vertical line on a 10-cm unstructured hedonic scale, with extremes defined as very undesirable to very desirable. The following parameters were evaluated: general appearance, aroma, taste, texture, and overall acceptability. Unsweetened tea was available to participants between samples.

### 2.9. Statistical Analysis

Data were statistically analysed using Statistica 13.1 software (StatSoft Inc., Tulsa, OK, USA). The selected statistical method was a one-way analysis of variance with Tukey’s test. A significance level of *p* < 0.05 was used. The results of the analyses are presented as the arithmetic mean along with the calculated standard deviation (SD).

## 3. Results and Discussion

### 3.1. Colour Measurement

Raw and heat-treated plant burgers were subjected to instrumental colour analysis in the CIE L* a* b* system (Table 2). Measurements were taken from the surface of the burgers. The burgers prepared according to the P-OO recipe, in their raw form, had the highest value of the L* (brightness) parameter (L = 52.81), while burgers from the P-OA group had the highest value after heat treatment (L = 40.35). Regardless of the plant fat and antioxidant ingredient used, the heat treatment process decreased the value of the L* parameter in all tested groups. The main ingredient in the burgers studied was textured pea protein, which made the proportion of the L* component of extruded pea protein (L = 49.14) in the study by Mandliya et al. [31] comparable to the results obtained in the present study. Similar results were obtained in the study by Xia et al. [32], where the extruded pea protein had a value of L* = 46.72. In the study by Botella-Martínez et al. [33], the soy protein-based burgers also had similar but slightly lower brightness (L = 43.74–44.47 raw; L = 34.72–34.87 cooked). The resulting brightness scores were higher for raw burgers than the Impossible™ brand plant-based patty samples analysed by Vu et al. [34]. However, for heat-treated burgers, the results obtained in this study were similar to those obtained by Vu et al. [34], where the L* value was 40.0.

The lowest value of the a* parameter (+a* red, −a* green) of the raw burger samples was characterised by samples belonging to the C-OO group (a = 12.96), a value that was statistically significantly different (*p* ≤ 0.05) from the rest of the control and olive oil samples, except for the H-OO group). Between the samples differing in the type of plant fat, regardless of the antioxidant ingredient used, the highest contribution of component a* was recorded for samples with acai oil. Heat treatment led to a decrease in the value of this component in all samples except the C-CO group. In the study by Botella-Martínez et al. [33], where beet juice was used to provide colour to soy protein-based burgers, the contribution of the a* component was 6.23–6.37 in raw burgers and 7.92–7.88 after heat treatment. These values are significantly lower than those obtained in the present study. The difference probably is due to the ingredients used and their proportion, including beet juice powder and some intensive red spices like sweet, hot, and smoked paprika. Significantly, raw burger samples analysed by Vu et al. [34] had similar values (a = 21.3) to those obtained in the present study. However, after heat treatment, the samples analysed by Vu et al. [34] had significantly lower a* component values (a = 8.0). The difference may be due to the use of soy leghemoglobin in Impossible™ brand products, the colour of which changes under heat treatment [20].

The highest contribution of the b*”comp’nent (+b* yellow; −b* blue) in raw burgers was observed in samples from the C-CO group (b = 28.03). This value was statistically significantly higher (*p* ≤ 0.05) than the other samples tested. In the literature data, the proportion of the b* component varies highly depending on the ingredients used and the technological process. Pea protein extruded in the study by Xia et al. [32] had a higher proportion of the b* component (b = 35.63), while in the study by Mandliya et al. [31] the value was 22.7. The results obtained in the present study fall within this range. The Impossible™ brand product studied by Vu et al. [34] had a b* value of 20.9 in raw form, which is lower than in the present study, but after frying, the b* component (b = 15.7) was close to one of the least yellow samples analysed in the present study (b = 13.84). As in the case of the L* and the a* components, the heat treatment process decreased the value of the b* parameter in all samples studied. Other studies on meat analogues have confirmed this observation, e.g., Vu et al. [34], where heat treatment reduced the contribution of the a* and the b* components in soy protein-based burgers.

### 3.2. Texture Measurement

The processed plant-based burgers were subjected to texture analysis, and the following parameters were examined: hardness, springiness, and cohesiveness (Figure 1 and Figure 2). It was shown that the samples from the control group had the lowest hardness, regardless of the type of plant fat used. It was found that the lowest hardness value was recorded in samples from the C-AO group (5.74 N). The highest hardness value was observed in samples with buckwheat honey (16.11–16.92). However, only the samples from the group containing olive oil and honey (16.98 N) differed among samples with the same plant fat in a statistically significant way (*p* ≤ 0.05). Plant fat had no statistically significant (*p* > 0.05) effect on hardness in this group.

Analysis of springiness showed no statistically significant (*p* > 0.05) differences between samples. The springiness value was in the range of 0.11–0.14. For cohesiveness, differences were observed between samples. The burgers with propolis extract and acai oil statistically were significantly (*p* ≤ 0.05) less cohesive (0.11). In contrast, the control samples with acai oil had the highest cohesiveness value (0.21).

In a study by Kaleda et al. (2021) [35], the hardness value of a burger based on pea and oat protein ranged from 4–81 N, indicating a wide variation in textural properties in meat analogues. Therefore, it is difficult to relate the obtained values to the literature data due to the variation in ingredients, proportions, and technological operations. Compared to the present study, higher values (13.5–27.33) were recorded in a meat analogue based on pea protein and oat protein in the study by de Angelis et al. (2020) [5]. However, both studies analysed a meat analogue in the form of extruded protein, unlike the present study, which examined a meat analogue in the form of a burger, in the recipe of which numerous ingredients affecting texture parameters (plant fat, pectin, carrageenan, spices) were present.

In the study by de Angelis et al. [5], higher springiness values (0.72–0.87) and cohesiveness (0.54–0.62) were also reported. Wi et al. [36] studied the effect of the proportion of liquid ingredients on the textural properties of a meat analogue based on hydrated textured plant protein. In this study, the pea protein-based burgers contained about 21% ingredients in liquid form, corresponding to the group with the addition of 30 g of liquid ingredients from the Wi et al. [36] study. The hardness results obtained by Wi et al. [36] were about 15 N, similar to most of the samples in the study groups of the present study. For cohesiveness, the values recorded by Wi et al. [36] were similar but slightly higher (around 0.22).

### 3.3. Fatty Acid Profile Analysis

The analysis indicated that the plant fat type significantly influenced the fatty acid profile of the examined samples (Table 3). Statistically significantly (*p* ≤ 0.05) higher levels of saturated fatty acids (SFA) characterised samples belonging to the H-OO group (10.70%). In each group characterised by the same antioxidant component and a different plant fat, a statistically significantly (*p* ≤ 0.05) higher level of SFA was observed for samples with olive oil. These results indicate a relationship with the sensitivity of polyunsaturated bonds to oxidation during thermal processing. In contrast, the lowest SFA value was reported in samples belonging to the JE-CO group (7.71%), indicating the ability of jalapeno extract to inhibit the formation of saturated fatty acids. This value statistically was not significantly different from samples in the H-CO group (7.96%). Research by Onopiuk et al. [25] confirms that jalapeno extract contains a high amount of phenolic compounds (mainly capsaicin), which can have a positive effect on fatty acid profile and can further reduce the formation of polycyclic aromatic hydrocarbons during heat treatment.

Analysis of monounsaturated fatty acid (MUFA) content showed that samples from the olive oil groups had the highest MUFA content (66.15–67.18%). Still, no statistically significant (*p* > 0.05) differences existed between samples with olive oil and various antioxidant ingredients. The lowest MUFA content among the groups differing in the type of plant fat was observed in samples with acai oil (55.20–56.64%). Thus, it was observed that the highest MUFA values were characteristic for samples containing olive oil, while the lowest was for samples with acai oil.

Regarding the sum of polyunsaturated fatty acids (PUFA), it was observed that meat analogues containing olive oil had the lowest PUFA content, and the differences between groups with the same plant fat in most groups were not statistically significant (*p* > 0.05). Among the groups, samples from the C-OO group contained the lowest amount of PUFA (22.47%). The highest statistically significant (*p* ≤ 0.05) PUFA content was reported for groups with acai oil (33.96–35.99%).

The highest PUFA/SFA ratio characterised groups with acai oil (3.62–4.26). This was a statistically significantly higher value (*p* ≤ 0.05) than the samples with the other plant fats. Only in the group with jalapeno extract were there no significant differences between acai oil and canola oil. The smallest PUFA/SFA ratio was observed in the groups containing olive oil, of which the lowest result was observed in the H-OO group (2.12).

The samples with olive oil contained statistically significantly (*p* ≤ 0.05) lower content of omega-3 and omega-6 fatty acids. The olive oil groups did not differ statistically significantly (*p* > 0.05), but the lowest values were recorded for the C-OO samples (omega-6 = 17.34%; omega-3 = 5.07%). Samples from the acai oil groups contained the highest amount of omega-6 fatty acids (28.2–30.171%), and they did not differ statistically significantly (*p* > 0.05). For omega-3 fatty acids, the highest values were found in the groups with canola oil (6.79–7.15%).

Analysing the n6/n3 ratio, the highest values were observed in the groups with acai oil (4.96–5.27). In contrast, the lowest n6/n3 ratio was found in the group with sunflower oil, regardless of the added extracts (2.80–3.08). According to a study by Siang et al. [37], individual plant oils differ significantly in their fatty acid profiles, especially canola oil and olive oil. The results are very similar to those obtained by Cui et al. [38], where canola oil contained 8.26% SFA, 62.22% MUFA, and 29.52% PUFA. Also, the study by Xin et al. [39] confirmed a similar fatty acid profile for rapeseed oil.

### 3.4. Determination of Phenolic Compounds, Anti-Inflammatory Activity, and DPPH Radical Scavenging Activity

Polyphenols and flavonoids are compounds with highly beneficial antioxidant and anti-inflammatory properties. They are also important as neuro- and cardio-protective, antiviral, and anticancer substances [40,41,42]. Plant-based meat analogues were subjected to analysis of the content of polyphenols, flavonoids, and anti-inflammatory activity (Table 4). The results showed that statistically significantly (*p* ≤ 0.05) higher content of polyphenols, regardless of the type of oil, was characterized by samples containing buckwheat honey. Among them, the H-AO group recorded the highest value (370.54 mg/100 g). A statistically significantly (*p* ≤ 0.05) lower amount of polyphenols was found in samples from the control group (129.15–145.36 mg/100 g). In a comparison between groups containing the same antioxidant ingredient but a different plant oil, it was observed that the use of canola oil resulted in a statistically significantly (*p* ≤ 0.05) lower polyphenol content than that of acai oil or olive oil. Samples containing acai oil contained statistically significantly higher polyphenol content than samples with canola oil and olive oil, except for the group with jalapeno extract, where the polyphenol content of the JE-OO and JE-AO groups was similar (239.83 mg/100 g; 245.72 mg/100 g, respectively).

Similar correlations were observed for the flavonoid content of the samples. Statistically significantly (*p* ≤ 0.05) lower values of flavonoids were present in the control group (72.27–77.77 mg/100 g), significantly higher in the groups with propolis and jalapeno peppers extract, and the highest in the groups with honey in the recipe (90.71–103.03). A comparison between groups containing different plant fats showed that canola oil led to a lower flavonoid content than olive oil or acai oil. There were no statistically significant differences between the groups with olive oil and acai oil (*p* > 0.05), except for the group with jalapeno pepper extract.

The anti-inflammatory activity (AI) of plant-based meat analogues was highly variable and ranged from 16–76 percent of inhibition. Statistically significantly (*p* ≤ 0.05) lower anti-inflammatory activity was found in samples belonging to the C-OO group (16%). For the study groups, it was observed that in samples containing the same antioxidant component, samples with acai oil had the lowest AI value. The statistically significantly highest (*p* ≤ 0.05) AI results were recorded in the C-CO (76%), P-OO (74%), and JE-OO (68%) samples. Propolis, as a mixture of tree resin and bee gland secretions, is also a source of anti-inflammatory compounds [43], so the burgers had higher levels of phenolic compounds and flavonoids than controls, which resulted in higher levels of phenolic compounds and flavonoids in the burgers compared to controls [43]. According to Yamaguchi et al. [44], phenolic compounds regulate free radicals formed during metabolism, inhibit the growth of microorganisms, and prolong the freshness and acceptability of products. The use of acai oil increased the content of phenolic compounds, including statistically significant flavonoids, since it is considered a “superfruit” and contains a lot of compounds that are secondary metabolites (it was identified to have about 90 bioactive compounds: 31% are flavonoids, 23% phenolic compounds, 9% anthocyanins, and 11% lignoids). Thanks to its composition, açai has proven antioxidant activity against hydroxyl, peroxyl, and 2,2-diphenyl picrylhydrazyl (DPPH) radicals, and it also inhibits liposome oxidation [45].

Plant-based meat analogues based on pea protein were tested for total antioxidant capacity using synthetic radical DPPH. Analysis of the results showed noticeable differences between the groups. In a comparison between groups containing the same antioxidant component, it was observed that the samples containing buckwheat honey showed the highest statistically significant (*p* ≤ 0.05) antioxidant capacity, regardless of the type of plant fat (83.07–88.44%). The statistically significantly lowest (*p* ≤ 0.05) free radical scavenging capacity was demonstrated using samples from the control group (40.69–51.33%) and the research group samples with jalapeno pepper extract (68.01–73.63%).

Analysis of the results in groups differing in the type of plant oil showed that samples with canola oil showed the lowest statistically significant (*p* ≤ 0.05) antioxidant capacity, while samples with acai oil showed the highest. The results of the total antioxidant capacity are coherent with the results of the analysis of the content of polyphenols and flavonoids in the samples. Ingredients that demonstrate antioxidant activity were used (the high free radical quenching ability of the components of honey, propolis and acai oil was confirmed) so that a significant increase in the antioxidant capacity of plant-based meat analogues was observed. Research by Hanula et al. [46] indicates that açai extract is a valuable source of polyphenols (31.36 ± 1220 mg gallic acid/g of sample) and has high antioxidant capacity (ABTS 50.54 ± 0.296 mg ascorbic acid/g of sample; FRAP 38.05 ± 1.268 mg ascorbic acid/g of sample).

### 3.5. Consumer Evaluation of Plant-Based Meat Analogues

The prepared burgers were subjected to semi-consumer analysis. The results are shown in Table 5. No statistically significant (*p* > 0.05) differences in overall appearance ratings between samples were observed. The highest values were observed for samples with acai oil (8.0–8.4). In terms of aroma, the evaluators showed no statistically significant differences between samples with different types of vegetable fat in the case of the group with honey. In the other groups, samples with canola oil scored the highest (7.4–8.9). Samples with jalapeno extract scored best for samples containing the same vegetable fat. However, they were not statistically significantly different (*p* > 0.05) from the samples in the control group for canola oil and acai oil.

The semi-consumer evaluation of the burgers showed no statistically significant (*p* > 0.05) effect of the type of vegetable fat on the taste of the product. The type of vegetable fat had a statistically significant (*p* ≥ 0.05) effect only on the texture of the burgers in the propolis extract group, where the burgers with acai oil scored lower compared to the other samples. For samples containing the same plant fat, samples with jalapeno extract were rated the highest in terms of general acceptability. However, they were not statistically significantly different (*p* > 0.05) from the samples in the control group for canola oil and acai oil.

## 4. Conclusions

Increasing consumer interest in meat analogues is affecting the dynamic development of this food category. The most important factor affecting the quality of a meat analogue is its formulation. Proper selection of product ingredients results in the desired physicochemical properties and high semi-consumer acceptability. Plant-based meat analogues based on textured pea protein with acai oil (AO), canola oil (CO), and olive oil (OO) were tested. Propolis extract (P), buckwheat honey (H) and jalapeno/bell pepper extract (JE) were added to the plant-based burger formulation. The texture and colour of the prepared plant-based burgers were examined, as well as selected chemical parameters. The results showed that the burgers from the control group were characterized by the lowest hardness, while the burgers with added buckwheat honey demonstrated the highest hardness. There was significant variation between samples in a fatty acid profile. The samples with olive oil had the highest monounsaturated fatty acid (MUFA) and saturated fatty acid (SFA) content. In contrast, the highest polyunsaturated fatty acid (PUFA) content was present in the samples with acai oil. The highest content of polyphenols and flavonoids in samples with honey also resulted in the highest antioxidant capacity of samples from this group. It was found that the complex matrix of the plant-based meat analogue resulted in varying effects of recipe changes on the parameters studied.

A semi-consumer analysis of the burgers was also carried out, which showed that the recipe changes had no significant effect for the evaluators on the overall appearance and, for most groups, on the taste and texture of the product. Samples with jalapeno extract had similar values to the control group in terms of most of the evaluated parameters. At the same time, burgers with jalapeno extract were rated highest regarding overall acceptability. Therefore, the study demonstrated the possibility of modifying the physicochemical properties of meat analogues through an appropriate selection of ingredients in the recipe with higher product acceptability.

## Figures and Tables

**Figure 1 foods-13-00071-f001:**
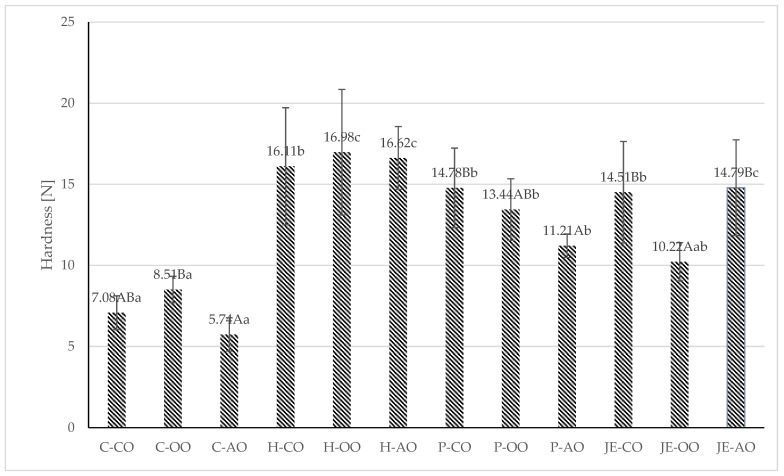
Hardness [N] of fried plant-based meat analogues. C—control group, H—honey group, P—propolis group, JE — jalapeno extract group, CO—canola oil, OO—olive oil, AO—acai oil; (A–C)—averages in columns with different letters show statistically significant differences between samples with the same antioxidant component, but different type of plant fat; (a–c)—averages in columns with different letters show statistically significant differences between samples with the same type of plant fat, but different antioxidant component; *p* ≤ 0.05.

**Figure 2 foods-13-00071-f002:**
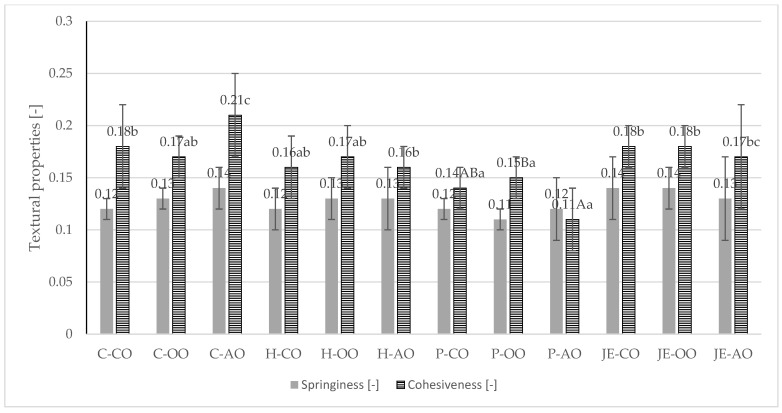
Springiness [-] and cohesiveness [-] of fried plant-based meat analogues. C—control group, H—honey group, P—propolis group, JE—jalapeno extract group, CO—canola oil, OO—olive oil, AO—acai oil; (A–C)—averages in columns with different letters show statistically significant differences between samples with the same antioxidant component, but different type of plant fat; (a–c)—averages in columns with different letters show statistically significant differences between samples with the same type of plant fat, but different antioxidant component; *p* ≤ 0.05.

**Table 1 foods-13-00071-t001:** Pea burger recipe composition.

Ingredient [g/100 g Burger]	C-CO	C-OO	C-AO	H-CO	H-OO	H-AO	H-CO	P-OO	P-AO	JE-CO	JE-OO	JE-AO
Pea extrudate	17.95	17.95	17.95	17.95	17.95	17.95	17.95	17.95	17.95	17.79	17.79	17.79
Water	61.3	61.3	61.3	55.32	55.32	55.32	55.32	55.32	55.32	60.61	60.61	60.61
Dried onion	10.77	10.77	10.77	10.77	10.77	10.77	10.77	10.77	10.77	10.77	10.77	10.77
Sweet paprika	0.18	0.18	0.18	0.18	0.18	0.18	0.18	0.18	0.18	0.18	0.18	0.18
Smoked paprika	0.18	0.18	0.18	0.18	0.18	0.18	0.18	0.18	0.18	0.18	0.18	0.18
Hot pepper	0.09	0.09	0.09	0.09	0.09	0.09	0.09	0.09	0.09	0.09	0.09	0.09
Bear garlic	0.39	0.39	0.39	0.39	0.39	0.39	0.39	0.39	0.39	0.39	0.39	0.39
Mushroom flavour	0.07	0.07	0.07	0.07	0.07	0.07	0.07	0.07	0.07	0.07	0.07	0.07
Yeast extract	0.04	0.04	0.04	0.04	0.04	0.04	0.04	0.04	0.04	0.04	0.04	0.04
Carrageenan	1.38	1.38	1.38	1.38	1.38	1.38	1.38	1.38	1.38	1.38	1.38	1.38
Pectin	1.38	1.38	1.38	1.38	1.38	1.38	1.38	1.38	1.38	1.38	1.38	1.38
Beet juice powder	0.29	0.29	0.29	0.29	0.29	0.29	0.29	0.29	0.29	0.29	0.29	0.29
Canola oil	5.98	0	0	5.98	0	0	5.98	0	0	5.98	0	0
Olive oil	0	5.98	0	0	5.98	0	0	5.98	0	0	5.98	0
Acai oil	0	0	5.98	0	0	5.98	0	0	5.98	0	0	5.98
Buckwheat honey	0	0	0	5.98	5.98	5.98	0	0	0	0	0	0
Propolis extract	0	0	0	0	0	0	5.98	5.98	5.98	0	0	0
Jalapeño extract	0	0	0	0	0	0	0	0	0	0.85	0.85	0.85

C—control group, H—honey group, P—propolis group, JE—jalapeno extract group, CO—canola oil, OO—olive oil, AO—acai oil.

**Table 2 foods-13-00071-t002:** CIE parameters (L*, a* and b*) in raw and fried plant-based meat analogues.

	Raw			Fried		
	L*	a*	b*	L*	a*	b*
C-CO	50.27 ± 0.81 ^Bab^	16.11 ± 0.61 ^Ba^	28.03 ± 0.7 ^Bb^	39.91 ± 0.81 ^Cb^	16.37 ± 0.14 ^Cb^	24.47 ± 0.90 ^Bc^
C-OO	48.46 ± 1.09 ^Aa^	12.96 ± 1.47 ^Aa^	24.14 ± 1.21 ^A^	34.99 ± 0.34 ^Aa^	11.56 ± 0.64 ^Aa^	18.39 ± 0.15 ^Aab^
C-AO	50.41 ± 1.48 ^Bab^	20.4 ± 0.15 ^Cb^	24.08 ± 0.24 ^A^	36.20 ± 0.53 ^Ba^	15.50 ± 0.29 ^Bbc^	18.25 ± 0.47 ^Abc^
H-CO	50.53 ± 1.27 ^ab^	16.92 ± 1.34 ^Bab^	24.48 ± 1.31 ^a^	34.26 ± 3.45 ^a^	12.13 ± 2.90 ^a^	13.84 ± 4.69 ^a^
H-OO	49.47 ± 2.36 ^a^	14.6 ± 1.84 ^Aab^	24.36 ± 2.24	36.21 ± 3.33 ^ab^	12.05 ± 3.68 ^a^	14.57 ± 5.13 ^a^
H-AO	48.25 ± 2.04 ^a^	17.68 ± 1.31 ^Ba^	23.87 ± 1.32	34.76 ± 3.48 ^a^	11.55 ± 2.57 ^a^	13.98 ± 4.75 ^a^
P-CO	51.54 ± 1.84 ^b^	17.69 ± 1.95 ^ab^	24.62 ± 1.45 ^a^	35.52 ± 2.25 ^Aa^	13.68 ± 2.01 ^Aab^	15.91 ± 2.43 ^Aa^
P-OO	52.81 ± 2.32 ^b^	17.88 ± 1.62 ^c^	25.54 ± 1.78	38.90 ± 2.16 ^Bb^	17.53 ± 1.90 ^Bb^	19.88 ± 1.88 ^Bb^
P-AO	51.68 ± 2.34 ^b^	19.28 ± 1.54 ^ab^	24.34 ± 1.73	40.35 ± 1.97 ^Bb^	17.59 ± 2.70 ^Bc^	21.03 ± 2.31 ^Bc^
JE-CO	49.68 ± 1.58 ^a^	18.43 ± 1.38 ^ABb^	25.76 ± 0.94 ^a^	38.71 ± 2.98 ^b^	15.21 ± 2.17 ^b^	19.00 ± 2.39 ^Bb^
JE-OO	49.94 ± 1.72 ^a^	17.72 ± 2.07 ^Abc^	24.71 ± 2.33	37.33 ± 2.81 ^ab^	15.59 ± 2.01 ^b^	17.92 ± 3.97 ^ABab^
JE-AO	50.51 ± 1.89 ^ab^	19.78 ± 2.26 ^Bb^	25.16 ± 2.13	35.85 ± 2.98 ^a^	13.72 ± 2.19 ^ab^	15.11 ± 2.78 ^Aab^

C—control group, H—honey group, P—propolis group, JEjalapeno extract group, CO—canola oil, OO—olive oil, AO—acai oil; (A–C)—averages in columns with different letters show statistically significant differences between samples with the same antioxidant component, but different type of plant fat; (a–c)—averages in columns with different letters show statistically significant differences between samples with the same type of plant fat, but different antioxidant component; *p* ≤ 0.05.

**Table 3 foods-13-00071-t003:** Fatty acid profile analysis of plant-based meat analogues.

	SFA	MUFA	PUFA	PUFA/SFA	n6	n3	n6/n3
C-CO	8.67 ± 0.05 ^Bc^	63.47 ± 0.48 ^Ba^	27.85 ± 0.53 ^B^	3.21 ± 0.08 ^Ba^	20.93 ± 0.55 ^B^	6.79 ± 0.02 ^Ca^	3.08 ± 0.09 ^Ab^
C-OO	10.33 ± 0.05 ^Cab^	67.18 ± 0.05 ^Cb^	22.47 ± 0.10 ^Aa^	2.18 ± 0.02 ^Aab^	17.34 ± 0.05 ^Aa^	5.07 ± 0.04 ^A^	3.42 ± 0.02 ^B^
C-AO	8.46 ± 0.00 ^Aa^	55.54 ± 0.01 ^A^	35.99 ± 0.03 ^C^	4.26 ± 0.02 ^Cb^	30.17 ± 0.00 ^C^	5.73 ± 0.00 ^Bb^	5.27 ± 0.00 ^Cab^
H-CO	7.96 ± 0.14 ^Aab^	65.00 ± 0.04 ^Bb^	27.03 ± 0.10 ^B^	3.40 ± 0.07 ^Bab^	19.83 ± 0.07 ^B^	7.09 ± 0.03 ^Cb^	2.80 ± 0.00 ^Aa^
H-OO	10.70 ± 0.04 ^Cc^	66.64 ± 0.05 ^Cab^	22.65 ± 0.10 ^Aa^	2.12 ± 0.02 ^Aa^	17.42 ± 0.01 ^Aa^	5.13 ± 0.09 ^A^	3.40 ± 0.03 ^B^
H-AO	9.61 ± 0.05 ^Bc^	55.20 ± 0.61 ^A^	35.18 ± 0.56 ^C^	3.66 ± 0.04 ^Ca^	29.61 ± 0.58 ^C^	5.45 ± 0.02 ^Ba^	5.43 ± 0.13 ^Cb^
P-CO	8.08 ± 0.01 ^Ab^	64.26 ± 0.03 ^Bab^	27.65 ± 0.04 ^B^	3.42 ± 0.01 ^Bab^	20.42 ± 0.03 ^B^	7.10 ± 0.00 ^Cb^	2.88 ± 0.00 ^Aa^
P-OO	10.47 ± 0.05 ^Cb^	66.15 ± 0.34 ^Ca^	23.37 ± 0.29 ^Ab^	2.23 ± 0.02 ^Abc^	18.13 ± 0.32 ^Ab^	5.14 ± 0.03 ^A^	3.53 ± 0.08 ^B^
P-AO	9.42 ± 0.01 ^Bb^	56.30 ± 0.06 ^A^	34.27 ± 0.05 ^C^	3.64 ± 0.00 ^Ca^	28.56 ± 0.05 ^C^	5.62 ± 0.00 ^Bb^	5.08 ± 0.01 ^Cab^
JE-CO	7.71 ± 0.03 ^Aa^	64.72 ± 0.15 ^Bb^	27.55 ± 0.12 ^B^	3.58 ± 0.00 ^Bb^	20.30 ± 0.07 ^A^	7.15 ± 0.05 ^Cb^	2.84 ± 0.01 ^Aa^
JE-OO	10.27 ± 0.01 ^Ca^	66.65 ± 0.05 ^Bab^	23.06 ± 0.05 ^Aab^	2.25 ± 0.01 ^Ac^	17.88 ± 0.13 ^Aab^	5.16 ± 0.03 ^A^	3.46 ± 0.05 ^B^
JE-AO	9.39 ± 0.03 ^Bb^	56.64 ± 1.17 ^A^	33.96 ± 1.21 ^C^	3.62 ± 0.14 ^Ba^	28.20 ± 1.14 ^B^	5.68 ± 0.06 ^Bb^	4.96 ± 0.15 ^Ca^

C—control group, H—honey group, P—propolis group, JE—jalapeno extract group, CO—canola oil, OO—olive oil, AO—acai oil; (A–C)—averages in columns with different letters show statistically significant differences between samples with the same antioxidant component, but different type of plant fat; (a–c)—averages in columns with different letters show statistically significant differences between samples with the same type of plant fat, but different antioxidant component; *p* ≤ 0.05.

**Table 4 foods-13-00071-t004:** Polyphenol and flavonoid content [mg/100 g], anti-inflammatory activity [%], and antioxidant activity [%] of samples of plant-based meat analogues.

	Polyphenols [mg GAE/100 g]	Flavonoids [mg Quercetin/100 g]	Anti-Inflammatory Activity [%]	DPPH Radical Scavenging Activity [%]
C-CO	129.15 ± 2.80 ^Aa^	72.27 ± 2.05 ^Aa^	76 ± 0.04 ^Cc^	40.69 ± 0.82 ^Aa^
C-OO	135.07 ± 1.83 ^Ba^	76.57 ± 0.56 ^Ba^	16 ± 0.02 ^Aa^	44.07 ± 0.84 ^Ba^
C-AO	145.36 ± 2.63 ^Ca^	77.77 ± 0.47 ^Ba^	51 ± 0.02 ^Bc^	51.33 ± 0.47 ^Ca^
H-CO	273.50 ± 4.42 ^Ac^	90.71 ± 2.93 ^Ac^	62 ± 0.02 ^Bb^	83.07 ± 1.03 ^Ad^
H-OO	306.78 ± 4.34 ^Bd^	97.14 ± 0.86 ^Bd^	58 ± 0.05 ^Ab^	86.58 ± 0.64 ^Bd^
H-AO	370.54 ± 3.99 ^Cd^	103.03 ± 0.52 ^Bd^	40 ± 0.06 ^Aab^	88.44 ± 0.51 ^Cd^
P-CO	228.12 ± 4.40 ^Ab^	88.46 ± 1.08 ^Ac^	63 ± 0.02 ^Bb^	72.24 ± 0.41 ^Ac^
P-OO	248.44 ± 4.15 ^Bc^	91.90 ± 2.39 ^Bc^	74 ± 0.02 ^Cc^	74.99 ± 0.30 ^Bc^
P-AO	264.64 ± 4.88 ^Cc^	96.09 ± 0.36 ^Bc^	50 ± 0.01 ^Abc^	79.58 ± 1.06 ^Cc^
JE-CO	225.34 ± 5.63 ^Ab^	81.91 ± 0.90 ^Ab^	52 ± 0.04 ^Ba^	68.01 ± 0.43 ^Ab^
JE-OO	239.83 ± 1.59 ^Bb^	85.70 ± 0.90 ^Bb^	68 ± 0.01 ^Cc^	70.67 ± 0.42 ^Bb^
JE-AO	245.72 ± 4.36 ^Bb^	89.71 ± 0.37 ^Cb^	31 ± 0.03 ^Aa^	73.63 ± 0.61 ^Cb^

C—control group, H—honey group, P—propolis group, JE—jalapeno extract group, CO—canola oil, OO—olive oil, AO—acai oil; (A–C)—averages in columns with different letters show statistically significant differences between samples with the same antioxidant component, but different type of plant fat; (a–d)—averages in columns with different letters show statistically significant differences between samples with the same type of plant fat, but different antioxidant component; *p* ≤ 0.05.

**Table 5 foods-13-00071-t005:** Semi-consumer scaling method of sensory evaluation of analyzed meat analogues.

Group	Overall Appearance	Aroma	Taste	Texture	Overall Acceptability
C-CO	8.0 ± 0.8	8.9 ± 1.1 ^Bb^	8.0 ± 1.1 ^b^	8.2 ± 0.7	8.2 ± 0.9 ^ab^
C-OO	8.1 ± 0.9	8.1 ± 1.1 ^ABb^	7.8 ± 0.8 ^bc^	8.0 ± 1.0	8.0 ± 0.8 ^b^
C-AO	8.2 ± 0.8	8.0 ± 0.5 ^Aa^	7.6 ± 0.6 ^b^	8.1 ± 0.7 ^b^	8.1 ± 0.6 ^bc^
H-CO	8.3 ± 0.9	8.0 ± 1.1 ^ab^	7.5 ± 0.6 ^b^	7.8 ± 0.9	7.8 ± 0.8 ^ab^
H-OO	7.9 ± 0.7	7.4 ± 0.5 ^ab^	7.1 ± 0.4 ^b^	7.6 ± 0.6	7.1 ± 0.7 ^a^
H-AO	8.3 ± 0.7	7.5 ± 0.5 ^a^	7.3 ± 0.4 ^b^	7.8 ± 0.6 ^b^	7.8 ± 0.9 ^b^
P-CO	8.0 ± 0.8	7.8 ± 0.7 ^Ba^	6.1 ± 0.5 ^a^	7.5 ± 0.6 ^B^	7.5 ± 0.9 ^a^
P-OO	7.9 ± 0.6	7.2 ± 0.5 ^Aa^	5.8 ± 0.5 ^a^	7.6 ± 0.6 ^B^	7.1 ± 0.7 ^a^
P-AO	8.0 ± 0.7	7.5 ± 0.6 ^ABa^	6.0 ± 0.4 ^a^	6.8 ± 0.7 ^Aa^	7.1 ± 0.4 ^a^
JE-CO	8.0 ± 0.6	8.9 ± 0.9 ^Bb^	8.2 ± 0.8 ^b^	7.8 ± 0.9	8.6 ± 0.7 ^b^
JE-OO	8.1 ± 0.9	8.0 ± 0.7 ^Ab^	8.0 ± 1.1 ^c^	7.5 ± 0.7	8.3 ± 0.6 ^b^
JE-AO	8.4 ± 0.7	8.6 ± 0.5 ^ABb^	7.8 ± 0.6 ^b^	7.8 ± 0.8 ^b^	8.5 ± 0.7 ^c^

C—control group, H—honey group, P—propolis group, JE—jalapeno extract group, CO—canola oil, OO—olive oil, AO—acai oil; (A–C)—averages in columns with different letters show statistically significant differences between samples with the same antioxidant component, but different type of plant fat; (a–c)—averages in columns with different letters show statistically significant differences between samples with the same type of plant fat, but different antioxidant component; *p* ≤ 0.05.

## Data Availability

Data is contained within the article.

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
