# Peer review of "The Effect of Type of Vegetable Fat and Addition of Antioxidant Components on the Physicochemical Properties of a Pea-Based Meat Analogue"

_foods, 2023, doi:10.3390/foods13010071_

Round 1

Reviewer 1 Report

Comments and Suggestions for Authors

General Comments:

In the absence of line numbers, it’s difficult to refer to the authors in this reviewer’s comments.

The authors provide an interesting paper on the potential impact of various vegetable oils on some physicochemical properties of pea protein-based meat analogue.  (Note the title should be changed to reflect pea protein vs other potential plant-based proteins.)  Also, note that the premise advanced by the authors was addressed decades ago by this reviewer.  More recently, the authors should consider the following relevant publications which were not cited:  Shen Y, Hong S, Li Y. Pea protein composition, functionality, modification, and food applications: A review. Adv Food Nutr Res. 2022;101:71-127. doi: 10.1016/bs.afnr.2022.02.002; Shanthakumar P, Klepacka J, Bains A, Chawla P, Dhull SB, Najda A. The Current Situation of Pea Protein and Its Application in the Food Industry. Molecules. 2022 Aug 22;27(16):5354. doi: 10.3390/molecules27165354;  

Abstract:

This simply presented section is reasonable, except the entire sentence that includes “high product acceptability” should be toned down.  One needs to consider exposure to “health-promoting compounds” and the health status of intended audiences before advancing the potential “health-promoting” effects.

Introduction:

The opening paragraph's condemnation of meat follows a popular mantra.  The authors should have provided some balance to their position, including the nutritional value of animal protein and the shortfall of meta-analogues for which the regulatory environment remains quite ambiguous, even in the USA.  A recent paper by Hertzler et al (Hertzler SR, Lieblein-Boff JC, Weiler M, Allgeier C. Plant Proteins: Assessing Their Nutritional Quality and Effects on Health and Physical Function. Nutrients. 2020 Nov 30;12(12):3704. doi: 10.3390/nu12123704) provides some valuable information to this point. 

While dietary patterns among consumers are dynamic, the nutritive value of these patterns are typically poor, and certainly not appropriate for underserved populations.

The authors should note that the protein from legumes is considered of low quality based on PDCAAS or even PER.  The statement that legume quality is similar to that of meat is without foundation.  At least the authors provide some evidence between plant and animal proteins on page 2.  What about potential allergenicity issues?

Even though some vegetable oils contain fat-soluble vitamins and possibly other potential nutrients, their contributions to the nutrient status of the consumer are minimal at best.  For example, unless one consumes “red” palm oil which contains lycopene, a precursor to vitamin A, the impact on global vitamin A status is not medically significant.

With the “anti-ingredient” movement among consumers (aka clean label), the addition of gums and emulsifiers to “meat” analogues is contrary to this movement.  At least the authors attempt to offer tangent remarks on clean label at the close of this section.  However, the emphasis on “natural” ingredients suggest that ingredients that are not considered natural via regulatory agencies are not safe…which is not true…the standards and definitions of which are not globally harmonized.

The brief introduction of analogue production via extrusion is interesting. 

The remarks on “natural antioxidants” are glib…what are the doses…which ones?

The antioxidant capacity is not standardized; in addition, none of these assessments is biologically relevant.  When addressing anti-inflammatory properties…the authors provide more glib remarks…there’s a need to quantify doses and nature of those compounds relative to maintaining a balance of physiological responses, such as those advanced by several types of cells, including T- and B-cells and an array of cytokines.

The analysis of polyphenols and flavonoids…while interesting… was not put in the context of physiological relevance.

Materials and Methods:

What assessments were made to determine that the fabricated burgers were safe for semi-consumer evaluation?  What does semi-consumer evaluation mean?

Color analysis via Lab is reasonable.

Texture assessment via Instron is reasonable.  How do these assessments compare with conventional burger products?

The approach for fatty acid analysis is reasonable.  Did the authors report the fat content per burger or RACC (reference amount customarily consumed)?

The methods for phenolics, flavonoids, and anti-inflammatory are reasonable.  Did the authors use any internal standards to assess effectiveness of extraction and to validate the methods for this artificial food matrix?

The DPPH method does not have any physiological significance.

The consumer evaluation approach is often used.  Were the panelists trained on how to use the 10 cm linear line?

Statistics were reasonable; the Statistica software is not any better than the stat tools available in Excel.

Results and Discussion:

The beet juice was the primary color source; would not expect any significant differences among the pea protein products.  The color analysis was overanalyzed.

Texture…would not expect differences among the products unless the saturated fat content differed, as suggested in table 4

Fatty acid analysis did not present any surprises.  The paper is not prepared to discuss controversial PAH “issues” or toxicology; the PUFA data are not surprising.

The polyphenol data are not surprising when considering the dynamics of the vegetable oil sources.

The reviewer’s experiences in sensory assessment suggest that the authors’ findings are not surprising.

Conclusions:

The addition of propolis extract, honey, and bell pepper do not augment nutritional value.  Propolis extract is associated with bee byproducts and potential allergens; honey has inconsistent composition; and bell peppers may provide additional vitamin C (ascorbic acid), yet the authors failed to assess the vitamin C or other potential nutrients in the finished products.  Thus, the phrases associated with nutritional value should be deleted.

The references to other physicochemical characteristics are consistent with the data provided by the authors.  The implied preference of the product with jalapeño extract is not surprising.  Jalapeño peppers, known for their pungency via the presence of capsaicin, are not surprising.  Did the authors attempt to match the “spiciness” across the products?

Realistically, the nutritional values across the products would not differ significantly; consumer acceptability can be modulated via selected or biased formulation.

Table 1

Formulation data; very difficult to read; please regenerate this table in landscape format

Table 2

Color analysis: difficult to read; please regenerate this table in landscape format

Table 3

Texture qualities; the key to the statistics is missing …although on the next page (9); publication needs to be sure the stats are at the table footnote

Table 4

Fatty acid analysis…

Tables 5 & 6 

No comments

Author Response

Reviewer 1

General Comments:

Comment 1: In the absence of line numbers, it’s difficult to refer to the authors in this reviewer’s comments.

Response: Line numbers have been added throughout the manuscript.

Comment 2: The authors provide an interesting paper on the potential impact of various vegetable oils on some physicochemical properties of pea protein-based meat analogue.  (Note the title should be changed to reflect pea protein vs other potential plant-based proteins.)  Also, note that the premise advanced by the authors was addressed decades ago by this reviewer.  More recently, the authors should consider the following relevant publications which were not cited: 

  • Shen Y, Hong S, Li Y. Pea protein composition, functionality, modification, and food applications: A review. Adv Food Nutr Res. 2022;101:71-127. doi: 10.1016/bs.afnr.2022.02.002;
  • Shanthakumar P, Klepacka J, Bains A, Chawla P, Dhull SB, Najda A. The Current Situation of Pea Protein and Its Application in the Food Industry. Molecules. 2022 Aug 22;27(16):5354. doi: 10.3390/molecules27165354;  

Response: The title of the manuscript have been changed to highlight pea protein:

“The effect of type of vegetable fat and addition of antioxidant components on the physicochemical properties of a plantpea-based meat analogue”

The authors also thank you for drawing attention to interesting publications in the area of pea protein. The authors have familiarized themselves with these publications and have included them in the references to the manuscript:

“Pea protein is the most promising plant protein for producing meat analogues in terms of consumer acceptance, availability, low allergenicity and technological properties such as water and oil holding capacity, gelation and solubility. This protein is valuable in terms of nutritional value, high lysine and threonine content and low glycemic index [14, 15]. The technological properties of plant proteins are highly variable and depend, among other things, on the method of protein isolation [15]. They can be modified to some extent through the use of appropriate chemical, physical or enzymatic processing [16].”

Abstract:

Comment 3: This simply presented section is reasonable, except the entire sentence that includes “high product acceptability” should be toned down.  One needs to consider exposure to “health-promoting compounds” and the health status of intended audiences before advancing the potential “health-promoting” effects.

Response: The authors agree with the reviewer's comment and have modified the abstract to mitigate the meaning.

“This study investigated the effect of the type of vegetable fat and health-promoting ingredient with antioxidant activity on the properties of meat analogue based on textured pea protein. “

“Therefore, it is possible to use selected ingredients with antioxidant activity.health-promoting compounds in the recipe of a plant-based burger with high product acceptability.”

Introduction:

Comment 4:

The opening paragraph's condemnation of meat follows a popular mantra.  The authors should have provided some balance to their position, including the nutritional value of animal protein and the shortfall of meta-analogues for which the regulatory environment remains quite ambiguous, even in the USA.  A recent paper by Hertzler et al (Hertzler SR, Lieblein-Boff JC, Weiler M, Allgeier C. Plant Proteins: Assessing Their Nutritional Quality and Effects on Health and Physical Function. Nutrients. 2020 Nov 30;12(12):3704. doi: 10.3390/nu12123704) provides some valuable information to this point. While dietary patterns among consumers are dynamic, the nutritive value of these patterns are typically poor, and certainly not appropriate for underserved populations.

Response: The authors emphasized in the opening paragraph the role of meat in providing complete protein in the human diet. They also thank the authors for suggesting an interesting study on the nutritional value of plant proteins.

“(…) investigate alternative sources of protein in the human diet [3, 5, 6]. Simultaneously, the significant role of meat in the human diet cannot be neglected, particularly in terms of providing highly digestible and complete protein [7].”

Comment 5: The authors should note that the protein from legumes is considered of low quality based on PDCAAS or even PER. The statement that legume quality is similar to that of meat is without foundation.  At least the authors provide some evidence between plant and animal proteins on page 2.  What about potential allergenicity issues?

Response: The authors agree with the reviewer's opinion that the functionality and nutritional value of legumes cannot be compared to meat. At the same time, the authors emphasize in the Introduction that the definition states that meat analogs should present similar nutritional value to the product they are supposed to replace.

“(…) is meat analogue. These products areshould be characterised by sensory properties and nutritional value similar to meat and meat products, so they cancould  play a function identical to meat in daily meals [9, 10].”

Nevertheless, the authors are aware that products available on the market do not always meet this assumption.

The manuscript also provides information on the PDCAAS index for selected plant proteins:

„Animal protein is characterised by high digestibility, full aminogram and specific tex-tural properties. In contrast, most plant proteins have a limiting amino acid and different functional properties. Among plant sources, soy protein (about 1.00), canola, potato, pea and quinoa protein (value at least 0.75) have the highest PDCAAS [7].”

As for the allergenicity of plant proteins, the authors briefly mention this aspect twice in the introduction:

“”However, soy and wheat are raw materials with high allergenic potential, and some consumers demonstrate a low level of tolerance for consuming soy or gluten products [13].”

Comment 6: Even though some vegetable oils contain fat-soluble vitamins and possibly other potential nutrients, their contributions to the nutrient status of the consumer are minimal at best.  For example, unless one consumes “red” palm oil which contains lycopene, a precursor to vitamin A, the impact on global vitamin A status is not medically significant.

Response:  The authors agree that fats added to the product contain marginal amounts of vitamins. In the manuscript, the phrase for plant fats had to mean that they can dissolve vitamins derived from other ingredients or added for fortification. This sentence has been rewritten to make its meaning clearer.

“Plant fat has a texturing role and carries flavour and fat-soluble vitamins. They also function as carriers of fat-soluble vitamins, which can be introduced into the product as a fortification or with other ingredients.”

Comment 7: With the “anti-ingredient” movement among consumers (aka clean label), the addition of gums and emulsifiers to “meat” analogues is contrary to this movement.  At least the authors attempt to offer tangent remarks on clean label at the close of this section.  However, the emphasis on “natural” ingredients suggest that ingredients that are not considered natural via regulatory agencies are not safe…which is not true…the standards and definitions of which are not globally harmonized.

Response: The authors agree that currently available meat analogs do not meet the objectives of "clean label" by the presence of gums and emulsifiers in the formulation. At the same time, the intention of the authors was not to criticize substances of synthetic origin, but only to note that consumers perceive these substances negatively. The relevant fragment in the manuscript:

“For this reason, several additives are used to protect the product and maintain its quality. The presence of such substances, such as those of synthetic origin, is perceived negatively by consumers who expect the so-called "clean label" [23]. Natural aAntioxidants of natural origin may be the answer to consumer expectations in this area. These include polyphenols and essential oils used as a product ingredient or packaging component.”

At the same time, the authors decided to rewrite some parts to reduce the emphasis on the phrase "natural”.

Natural aAntioxidants of natural origin may be the answer to consumer expectations in this area. These include polyphenols and essential oils used as a product ingredient or packaging component. The possibility of using natural such ingredients: and extracts from pepper (Piper nigrum L.), oregano (Origanum vulgare L.) juniper, (Juniperus communis L.), jalapeño extract, as well as catuaba, galangal and honey to extend the shelf life of meat products has been studied [24-27] (Munekata et al., 2020; Onopiuk et al., 2022; Półtorak et al., 2019; Tomović et al., 2020).”

“Selected natural ingredients with antioxidant activity were used to formulate plant-based burgers.”

Comment 8: The brief introduction of analogue production via extrusion is interesting.

Response: The authors thank you very much for your comment.

Comment 9: The remarks on “natural antioxidants” are glib…what are the doses…which ones?

Response: The authors meant ingredients of natural origin (e.g.: oregano extract, pepper extract), which contain substances with antioxidant effects such as: catechins, phenols, anthocyanins, etc.

Comment 10: The antioxidant capacity is not standardized; in addition, none of these assessments is biologically relevant.  When addressing anti-inflammatory properties…the authors provide more glib remarks…there’s a need to quantify doses and nature of those compounds relative to maintaining a balance of physiological responses, such as those advanced by several types of cells, including T- and B-cells and an array of cytokines.

Response: The analytical methods used to determine antioxidant (using DPPH, ABTS and FRAP radicals) and anti-inflammatory properties are commonly used by other researchers. These methods aim to quantify these properties to allow comparison with other results. Directly placing the results obtained in a physiological context is indeed not possible. Nevertheless, these methods are well-known and widely used in food analytics. They give an overall view of the properties of a food product. The research methods were chosen based on a detailed literature review and the authors' experience. Examples of publications with analysis using the DPPH radical:

Abdullah, F. A. A., Dordevic, D., Kabourkova, E., Zemancová, J., & Dordevic, S. (2022). Antioxidant and Sensorial Properties: Meat Analogues versus Conventional Meat Products. Processes, 10(9), 1864. doi.org/10.3390/pr10091864

Jeon, Y. H., Gu, B. J., & Ryu, G. H. (2023). Investigating the Potential of Full-Fat Soy as an Alternative Ingredient in the Manufacture of Low-and High-Moisture Meat Analogs. Foods, 12(5), 1011.  doi.org/10.3390/foods12051011

Cho, S. Y., & Ryu, G. H. (2021). Effects of mealworm larva composition and selected process parameters on the physicochemical properties of extruded meat analog. Food Science & Nutrition, 9(8), 4408-4419.

 doi.org/10.1002/fsn3.2414

Szpicer, A., Onopiuk, A., Barczak, M., & Kurek, M. (2022). The optimization of a gluten-free and soy-free plant-based meat analogue recipe enriched with anthocyanins microcapsules. LWT, 168, 113849.

 doi.org/10.1016/j.lwt.2022.113849

Comment 11: The analysis of polyphenols and flavonoids…while interesting… was not put in the context of physiological relevance.

Response: The authors enhanced the manuscript with information related to the properties of polyphenols and flavonoids in a physiological context.

“3.4. Determination of phenolic compounds, anti-inflammatory activity and DPPH radical scavenging activity

Polyphenols and flavonoids are compounds with highly beneficial antioxidant and anti-inflammatory properties. They are also important as neuro- and cardio-protective, antiviral and anticancer substances [40-42]. Plant-based meat analogues were subjected to analysis of the content of polyphenols, flavonoidand anti-inflammatory activity (Table 5).“

Materials and Methods:

Comment 12:  What assessments were made to determine that the fabricated burgers were safe for semi-consumer evaluation?  What does semi-consumer evaluation mean?

Response: Some of the ingredients used are widely available for purchase by consumers in grocery stores. They were bought new and unopened. Ingredients not available to retail customers were also approved for consumption and were obtained from trusted suppliers with quality specifications certifying their high quality. All ingredients were within their expiration dates. The preparation of the burgers was carried out with the highest standards of hygiene in the university's food preparation area. Storage and thermal processing of the burgers was carried out with the strictest care and in a manner that ensured consumer safety. The experiment, including the evaluation of the burgers by consumers, was carried out with the approval of the Ethics Committee for research involving human subjects at the Institute of Human Nutrition Sciences of the Warsaw University of Life Sciences, after the submission of an appropriate petition.

Semi-consumer evaluation is a type of hedonic consumer analysis. Non-trained consumers participate in such a study, and do not take part in analytical testing. Information is obtained on the basis of a properly prepared questionnaire, which the consumer is asked to fill out. The division into consumer and semi-consumer evaluation is based on the number of participants. Consumer analysis usually involves more than 100 participants, while semi-consumer evaluation usually involves 30-40 participants.

Comment 13: Color analysis via Lab is reasonable.

Response: The authors thank you very much for your comment.

Comment 14: Texture assessment via Instron is reasonable.  How do these assessments compare with conventional burger products?

Response: The developed burgers had a significantly lower hardness than conventional meat burgers. The authors believe that further research is needed to improve the texture of the plant-based meat analogue. The article compares the texture of the plant-based burger analogue with other similar products based on research by other scientists.

Comment 15: The approach for fatty acid analysis is reasonable.  Did the authors report the fat content per burger or RACC (reference amount customarily consumed)?

Response: The content of the raw material - vegetable fat (Canola, acai or olive oil) was reported per 100 g of burger. While the fatty acid profile is presented per 100 g of fat (the unit [%] was used).

“Fatty acids were identified by comparing the retention times observed with those of fatty acid standards (SupelcoTM 37 Component FAME mix, Sigma, St. Louis, MO, USA). The results were presented as an arithmetic mean and expressed in grams per 100 g of fat.”

Comment 16: The methods for phenolics, flavonoids, and anti-inflammatory are reasonable.  Did the authors use any internal standards to assess effectiveness of extraction and to validate the methods for this artificial food matrix?

Response: The methods used have been used for similar matrices but have not been validated for this particular matrix. For this reason, the analyses were performed in independent replicates.

Comment 17: The DPPH method does not have any physiological significance.

Response: The DPPH radical is a synthetic reagent for determining antioxidant capacity. We agree that it has no physiological science, but in developing the study we decided to apply a widely used analytical method using it. As a result, we determined the antioxidant capacity of the product in relation to this radical. Antioxidant capacity, on the other hand, has physiological significance.

Comment 18: The consumer evaluation approach is often used.  Were the panelists trained on how to use the 10 cm linear line?

Response: The consumer evaluation involves participants who are untrained in professional sensory analysis. However, each person taking part in the evaluation was instructed on how to fill out a questionnaire containing linear scales. The authors made sure that the instructions and task were understood by the evaluators.

Comment 19: Statistics were reasonable; the Statistica software is not any better than the stat tools available in Excel.

Response: The authors thank you very much for your comment.

Results and Discussion:

Comment 20: The beet juice was the primary color source; would not expect any significant differences among the pea protein products.  The color analysis was overanalyzed.

Response: The description of the color analysis results has been significantly shortened. Thank you for your valuable comment.

“Raw and heat-treated plant burgers were subjected to instrumental colour analy-sis in the CIE L*a*b* system (Table 2). Measurements were made on the surface of the burgers. The burgers prepared according to the P-OO recipe, in their raw form, had the highest value of the L* (brightness) parameter (L=52.81), while burgers from the P-OA group had the highest value after heat treatment (L=40.35). Among the non-heated samples, samples from the H-AO group had the lowest brightness (48.25). As for the heat-treated samples, the lowest value of the L* parameter was recorded in the sam-ples from the group with honey and canola oil (34.26). This value was statistically not significantly different (p>0.05) from the samples with the same antioxidant component but a different oil. Regardless of the plant fat and antioxidant ingredient used, the heat treatment process decreased the value of the L* parameter in all tested groups..”

“The lowest value of the a* parameter (+a*red, -a*green) of the raw burger samples was characterised by samples belonging to the C-OO group (a=12.96), a value that was statistically significantly different (p ≤ 0.05) from the rest of the control and olive oil samples, except for the H-OO group). In contrast, the highest contribution of the a* com-ponent was observed in samples from the C-AO (20.40), J-AO (19.78) and P-AO (19.28) groups, which were statistically not significantly different (p > 0.05). Between the samples differing in the type of plant fat, regardless of the antioxidant ingredient used, the highest contribution of component a* was recorded for samples with acai oil. The highest contribution of the a* component among the heat-treated burgers was record-ed for a sample from the P-AO group (17.59). Still, this value was statistically not significantly different (p > 0.05) from that recorded for samples from the P-OO (17.53) and C-AO (15.50) groups. The lowest value of the a* parameter among the samples after heat treatment was characterised by samples belonging to the group with honey and acai oil (11.55). In the case of the a* component, the hHeat treatment process led to a decrease in the value of this component in all samples except the C-CO group..”

“The highest contribution of the b* component (+b* yellow; -b* blue) in raw burgers was observed in samples from the C-CO group (b=28.03). This value was statistically significantly higher (p ≤ 0.05) than the other samples tested. The smallest level of yellow was found in burgers from the group containing honey and acai oil (H-AO = 23.87). Still, there were no significant statistical differences (p > 0.05) compared to the other groups. Among the heat-treated burger samples, the highest contribution of the a* component was recorded for samples from the C-CO group (24.47) and the lowest for the H-CO samples (13.84). The smallest level of yellow was found in burgers from the group containing honey and acai oil (H-AO = 23.87). Still, there were no significant statistical differences (p > 0.05) compared to the other groups. Among the heat-treated burger samples, the highest contribution of the a* component was recorded for samples from the C-CO group (24.47) and the lowest for the H-CO samples (13.84).

Comment 21: Texture…would not expect differences among the products unless the saturated fat content differed, as suggested in table 3. Fatty acid analysis did not present any surprises.  The paper is not prepared to discuss controversial PAH “issues” or toxicology; the PUFA data are not surprising. The polyphenol data are not surprising when considering the dynamics of the vegetable oil sources.  The reviewer’s experiences in sensory assessment suggest that the authors’ findings are not surprising.

Response: The authors thank you for your valuable consideration and appreciate the reviewer's great experience. The research conducted had a basic character, it was designed to provide systematic data in the studied area, and in this context the objective was achieved.

Conclusions:

Comment 22: The addition of propolis extract, honey, and bell pepper do not augment nutritional value.  Propolis extract is associated with bee byproducts and potential allergens; honey has inconsistent composition; and bell peppers may provide additional vitamin C (ascorbic acid), yet the authors failed to assess the vitamin C or other potential nutrients in the finished products.  Thus, the phrases associated with nutritional value should be deleted.

Response: The phrases associated with nutritional value have been removed.

“Propolis extract (P), buckwheat honey (H) and jalapeno bell pepper extract (JE) were added to the plant-based burger formulation to improve its nutritional value.”

“Therefore, the study demonstrated the possibility of modifying the physicochemical properties and improving the nutritional value of meat analogues through an appropriate selection of ingredients in the recipe with higher product acceptability”

Comment 23: The references to other physicochemical characteristics are consistent with the data provided by the authors.  The implied preference of the product with jalapeño extract is not surprising.  Jalapeño peppers, known for their pungency via the presence of capsaicin, are not surprising.  Did the authors attempt to match the “spiciness” across the products?

Response: The authors did not attempt to adjust the "spiciness" of the products, as it would then also be necessary to balance the "sweetness" coming from the honey. The authors concluded that making such changes to the recipe would introduce too many variables into the analysis.

Comment 24: Realistically, the nutritional values across the products would not differ significantly; consumer acceptability can be modulated via selected or biased formulation.

Response: The semi-consumer evaluation was intended to provide information on the impact of the burger recipe modifications performed on product acceptability. Consumers evaluated the samples in a randomized manner. The samples were also coded so that consumers would not make a biased evaluation.

Tables and figures

Comment 25:Table 1 - Formulation data; very difficult to read; please regenerate this table in landscape format.

Response: The table has been edited. The data is presented in the table in horizontal orientation.

Comment 26: Table 2 - Color analysis: difficult to read; please regenerate this table in landscape format.

Response: The table has been edited. The data is presented in the table in horizontal orientation.

Comment 27: Table 3 - Texture qualities; the key to the statistics is missing …although on the next page (9); publication needs to be sure the stats are at the table footnote.

Response: The data from Table 3, in accordance with the comment of one of the reviewers, was presented in the form of a graph. A key to the statistics was added directly below it.

Comment 28: Table 4 - Fatty acid analysis…

Response: The title of the table was completed with the missing word.

Table 3. Fatty acids profile analysis of plant-based meat analogues.

Reviewer 2

Comment 1: This study gives a positive understanding about the vegetarian food in burgers. Since the extremely hard work, can we have the burger figures to see the results? To see which one is hardness? The author can give the reader a visual picture is more convincing.

Response: In accordance with the reviewer's comment, the authors decided to present data on textural properties in visual form.

Comment 2: Why do the author choose acai oil (AO), canola oil (CO) and olive oil (OO), propolis extract (P), buckwheat honey (H) and jalapeno pepper extract (JE) for further analysis? These are not only vegetable oil, but some polyphenol substance. Why compare them together? Since the polyphenol substance may have more antioxidant capacities.

Response: The study design was to analyze the effect of two independent variables: oil (canola, acai, oil) and an antioxidant component (honey, propolis extract and jalapeno extract) on the properties of the burgers. Thus, the authors' objective was not to compare the antioxidant properties of oil to, for example: honey.

Comment 3: The key words maybe unless 6 words.

Response: Keywords have been reduced to 6.

Comment 4: There is no Line numbers, Table 2 and 3 need some modifications with standard deviation (±).

Response: Line numbers have been added throughout the manuscript. A plus-minus sign was added in place of the wrong sign in all tables.

Comment 5: The whole references need to match this journal.

Response: The format of all references has been corrected.

Comment 6: 3.5 Consumer evaluation of plant-based meat analogues, in this part, with no significant difference do not need to make emphasis.

Response: The description of the results of the semi-consumer evaluation has been significantly revised.

“The semi-consumer evaluation of the burgers showed no statistically significant (p > 0.05) effect of the type of vegetable fat on the taste of the product. The type of vegetable fat had a statistically significant (p ≥ 0.05) effect only on the texture of the burgers in the propolis extract group, where the burgers with acai oil scored lower compared to the other samples. For samples containing the same plant fat, samples with jalapeno extract were rated the highest in terms of general acceptability. However, they were not statistically significantly different (p > 0.05) from the samples in the control group for canola oil and acai oil.”

Reviewer 3

Comment 1: Space

Response: The unnecessary space has been removed.

“Reducing or excluding meat from the diet is a decision made by consumers as a result of growing dietary awareness, concern for the environment and the availability of key resources, biodiversity and animal welfare [1]. “

Comment 2: point

Response: The missing point was added at the end of the sentence.

“However, soy and wheat are raw materials with high allergenic potential, and some consumers demonstrate a low level of tolerance for consuming soy or gluten products [13].”

Comment 3: It is not appropriate format for references. Please use [14-16]. Revise the whole text.

Response: The format of references has been corrected.

“Solid fats extracted from coconut and cocoa are used to achieve a marbling effect, while other fats (i.e., sesame oil, avocado oil) are used to improve the fatty acid profile and flavour of the product [17-19]. Many other ingredients create the formulation of the meat analogue depending on the expected characteristics of the product, including pectins, polysaccharide gums, extracts with colouring activity, herbs, spices, yeast extract, nucleotides, sugar, enzymes and antioxidants [12, 18-20].”

Comment 4: Include numbers for these references.

Response: The format of references has been corrected the appropriate numbers have been added in the text.

“The possibility of using natural such ingredients: and extracts from pepper (Piper nigrum L.), oregano (Origanum vulgare L.) juniper, (Juniperus communis L.), jalapeño ex-tract, as well as catuaba, galangal and honey to extend the shelf life of meat products has been studied [24-27] (Munekata et al., 2020; Onopiuk et al., 2022; Półtorak et al., 2019; Tomović et al., 2020).”

Comment 5: The quality of Table 1 is poor. Please fix it for better readability.

Response: The table has been edited. The data is presented in the table in horizontal orientation.

Comment 6: In results you present only one replicate.

Response: Yes, the analysis was performed for 3 replicates for each sample. In the results, the average values for these 3 repetitions were presented, which is why there is one value for each sample.

“The analysis was performed in 3 replicates. Fatty acids were identified by comparing the retention times observed with those of fatty acid standards (SupelcoTM 37 Component FAME mix, Sigma, St. Louis, MO, USA). The results were presented as an arithmetic mean and expressed in grams per 100 g of fat.”

Comment 7: which standard compound? Rutin? Quercetin? Catechine?

 Response: The compound used to prepare the standard curve was quercetin.

“The results were expressed as the arithmetic mean of the three measurements in mg of quercetin per 100 g sample.”

This information was also added in the sentence about the standard curve.

“The total flavonoid content was determined from the standard curve for quercetin.”

Comment 8: ..for all L*a*b* parameters in whole text ..

Response:  Thank you for your valuable observation. The notations have been pored over throughout the manuscript.

“he burgers prepared according to the P-OO recipe, in their raw form, had the highest value of the L* (brightness) parameter (L=52.81), while burgers from the P-OA group had the highest value after heat treatment (L=40.35).”

“The lowest value of the a* parameter (+a*red, -a*green) of the raw burger samples was characterised by samples belonging to the C-OO group (a=12.96), a value that was statistically significantly different (p ≤ 0.05) from the rest of the control and olive oil samples, except for the H-OO group).”

“The highest contribution of the b* component (+b* yellow; -b* blue) in raw burgers was observed in samples from the C-CO group (b=28.03).”

Comment 9: Reference number. This citation error can be observed throughout the work. Please correct it.

Response: The format of references has been corrected.

Comment 10: L=46.72

Response: The sentence has been corrected.

Similar results were obtained in the study by Xia et al. [32], where the extruded pea protein had a value of L*=46.7272an L* component of 46.72.

Comment 11: the plus minus sign is omitted.

Response: A plus-minus sign was added in place of the wrong sign in all tables.

Comment 12: in row or column you need to emphasize.

Response: Statistical differences apply to data within a column, because each column represents the result for a different tested parameter. This is clarified in the explanatory notes to the tables.

“C - control group, H - honey group, P - propolis group, JE - jalapeno extract group, CO - canola oil, OO - olive oil, AO - acai oil; (A-C) - averages in columns with different letters show statistically significant differences between samples with the same antioxidant component, but different type of plant fat; (a-c) - averages in columns with different letters show statistically significant differences between samples with the same type of plant fat, but different antioxidant component; p ≤ 0.05.”

Comment 13:  Where is SD? 

Response: The authors thank you for your valuable comment and apologize for this missing information. SD has been added to the table.

Comment 14: Where is antioxidant activity in the title of Table 5?

Response: The title of the table has been changed to include antioxidant activity.

Table 5. Polyphenol and flavonoid content [mg/100g], and anti-inflammatory activity [%] and antioxidant activity [%] of samples of plant-based meat analogues.

Comment 15: Please present the results as mgGAE/100g.

Response: The results are presented as mg GAE/100 g. The caption in the table header was inadequate. It has been completed.

Comment 16: The same comment for flavonoids when you add standard compound.

Response: Results are shown as mg quercetin/100 g. The caption in the table header was inadequate. It has been completed.

Comment 17: Where is SD?

Response: The authors thank you for your valuable comment and apologize for this missing information. SD has been added to the table.

Comment 18: Please include author contribution.

Response: Authors contributions has been added to the manuscript.

Conceptualization, Anna Onopiuk and Andrzej Półtorak; Methodology, Klaudia Kołodziejczak; Software, Arkadiusz Szpicer; Validation, Anna Onopiuk; Formal analysis, Andrzej Półtorak; Investigation, Klaudia Kołodziejczak and Anna Onopiuk; Data curation, Klaudia Kołodziejczak and Arkadiusz Szpicer; Writing – original draft, Klaudia Kołodziejczak; Writing – review & editing, Anna Onopiuk and Andrzej Półtorak; Visualization, Arkadiusz Szpicer; Supervision, Andrzej Półtorak.

Comment 19: Some corrections are needed to ensure that the format of writing is done in accordance with the template of this Journal, including the format of the references and the title.

Response: The format of all references has been corrected.

Reviewer 2 Report

Comments and Suggestions for Authors

This study gives a positive understanding about the vegetarian food in burgers. Since the extremely hard work, can we have the burger figures to see the results? To see which one is hardness? The author can give the reader a visual picture is more convincing.

This study focus on the texture, colour and selected chemical parameters of plant-based burgers. It is a very systematic study. But I still have some unclear questions to ask.

1. Why do the author choose acai oil (AO), canola oil (CO) and olive oil (OO), propolis extract (P), buckwheat honey (H) and jalapeno pepper extract (JE) for further analysis? These are not only vegetable oil, but some polyphenol substance. Why compare them together? Since the polyphenol substance may have more antioxidant capacities.

2. The key words maybe unless 6 words.

3. There is no Line numbers, Table 2 and 3 need some modifications with standard deviation (±).

4. The whole references need to match this journal.

5. 3.5 Consumer evaluation of plant-based meat analogues, in this part, with no significant difference do not need to make emphasis.

Comments on the Quality of English Language

minor revision

Author Response

(The authors gave the same response as above.)

Reviewer 3 Report

Comments and Suggestions for Authors

Dear,

I carefully read the manuscript entitled: "The effect of type of vegetable fat and addition of antioxidant components on the physicochemical properties of a plant-based meat analogue". The paper has a lot of technical defects, therefore, major corrections are needed before further consideration. My comments are shown in a pdf file. 

Kind regards.

Author Response

(The authors gave the same response as above.)

Round 2

Reviewer 2 Report

Comments and Suggestions for Authors

Good!

Comments on the Quality of English Language

Good!